# Query2GMM: Learning Representation with Gaussian Mixture Model for Reasoning over Knowledge Graphs

## ABSTRACT

Logical query answering over Knowledge Graphs (KGs) is a fundamental yet complex task. A promising approach to achieve this is to embed queries and entities jointly into the same embedding space. Research along this line suggests that using multi-modal distribution to represent answer entities is more suitable than uni-modal distribution, as a single query may contain multiple disjoint answer subsets due to the compositional nature of multi-hop queries and the varying latent semantics of relations. However, existing methods based on multi-modal distribution roughly represent each subset without capturing its accurate cardinality, or even degenerate into uni-modal distribution learning during the reasoning process due to the lack of an effective similarity measure. To better model queries with diversified answers, we propose Query2GMM for answering logical queries over knowledge graphs. In Query2GMM, we present the GMM embedding to represent each query using a univariate Gaussian Mixture Model (GMM). Each subset of a query is encoded by its cardinality, semantic center and dispersion degree, allowing for precise representation of multiple subsets. Then we design specific neural networks for each operator to handle the inherent complexity that comes with multi-modal distribution while alleviating the cascading errors. Last, we design a new similarity measure to assess the relationships between an entity and a query's multi-answer subsets, enabling effective multi-modal distribution learning for reasoning. Comprehensive experimental results show that Query2GMM outperforms the best competitor by an absolute average of 6.35%.

## CCS CONCEPTS

• **Computing methodologies → Probabilistic reasoning**;

## KEYWORDS

Knowledge Graph, Probabilistic Reasoning, Logical Query, Multi-modal Distribution, Neural Reasoning

**ACM Reference Format:**
Anonymous Author(s). 2024. Query2GMM: Learning Representation with Gaussian Mixture Model for Reasoning over Knowledge Graphs. In *Proceedings of Make sure to enter the correct conference title from your rights confirmation emai (WWW'24).* ACM, New York, NY, USA, 10 pages. https://doi.org/XXXXXXX.XXXXXXX

## 1 INTRODUCTION

Knowledge Graphs (KGs) are structured heterogeneous graphs, which are constructed based on entity-relation-entity triplets. A fundamental task in knowledge graph reasoning, known as *logical query answering*, involves answering First-Order Logic (FOL) queries over KGs using operators including existential quantification ($\exists$), conjunction ($\wedge$), disjunction ($\vee$) and negation ($\neg$). This has wide real-life applications, such as web search [22, 36] and medical research [9, 23], among which real-life KGs often exhibit incompleteness according to the Open World Assumption [20]. To handle such incompleteness, numerous efforts have been devoted to developing neural models for complex logical query answering [1, 5, 10, 17–19, 29, 38, 41, 48, 51, 54].

Along this line, most existing approaches for logical reasoning over KGs work under the assumption that answer entities of each query typically follow a uni-modal distribution [9, 10, 37, 38, 54]. Based on this assumption, they design various embedding backbones based on geometric structures [10, 25, 33, 37, 54], probabilistic distributions [9, 19, 38, 52] and other basis [18, 41]. These structures are used to embed logical queries and answer entities into the same embedding space, where all answer entities of a query are expected to be encompassed within a large single region. For example, Q2B [37] develops the box embedding for each query and the point embedding for each entity, assuming that all answer entities for a query are inside or close to the box region corresponding to that query. PERM [9] applies a multivariate Gaussian distribution to encode the semantic location and spatial query area for each query's answers with a soft boundary, following the same assumption. However, this broad assumption compromises the expressiveness of logical queries and entities, as it inevitably includes many false positives in the large single region for a query.

Recently, research has found that the ideal query embedding may follow a multi-modal distribution [1] in the embedding space due to the compositional nature of multi-hop queries [5]; additionally, a relation can have multiple unknown latent semantics for different concrete triples [45]. For instance, given a logical query "Who has been nominated for Emmy Award?", the relation "Award Nominated" has several latent semantics including Actor Award, Artist Award, Actress Award, etc. Consequently, the intermediate entity set for individuals nominated for the Emmy Award may vary in terms of gender, nationality and other factors, leading to answer entities forming multiple disjoint subsets, as shown in Fig. 1a. A few recent studies aim to model the diversity of answer entities by designing appropriate embedding backbones that encode multiple disjoint subsets of answer entities. Specifically, NMP-QEM [29] leverages multivariate Gaussian mixture distribution to encode multiple disjoint answer subsets of each query, but it has to transform the Gaussian mixture distribution into a single centroid vector when

---

[1]A multi-modal distribution has two or more distinct peaks in its probability density function.

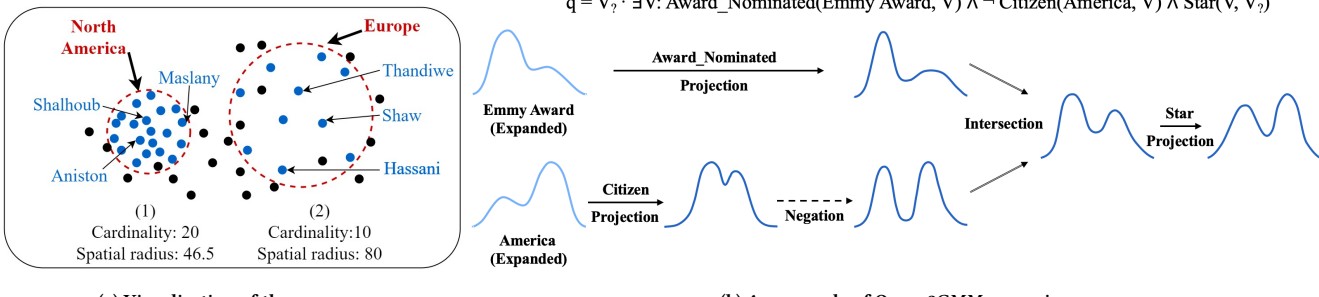

q = V$_?$ · ∃V: Award_Nominated(Emmy Award, V) ∧ ¬ Citizen(America, V) ∧ Star(V, V$_?$)

(a) Visualization of the answers.

(b) An example of Query2GMM reasoning.

**Figure 1: (a) Visualization of the answer distribution of the query "Who has been nominated for Emmy Award?" (b) Illustration of reasoning of Query2GMM on the query "What are the movies starring non-American actors who have been nominated for Emmy Award?" (2-modal distribution).**

measuring relationships between entities and queries in the embedding space. As a result, NMP-QEM degenerates to uni-modal distribution learning during the reasoning process. Query2Particles [5] proposes to encode each query with a set of vectors (particles), where each vector represents an answer subset according to its setting. However, this is problematic that it neglects the cardinality of these subsets, leading to an ambiguous answer entity set and compromising the performance of query answering.

Despite the potential advantages of multi-modal distribution modelling, existing methods struggle to learn accurate subset representation and even fail to assess the relationships between entities and queries in the multi-modal distribution context, limiting and violating the expressiveness of the query embedding. To further improve embedding quality, this presents two primary challenges: **Challenge I: How to accurately quantify different answer subsets of each query?** In practice, answer entities to each query may contain multiple disjoint subsets. Different from the uni-modal distribution, it is essential to accurately characterize the semantic center and cardinality of each subset in the multi-modal distribution context, which can provide prompts for reasoning and answer retrieval. Existing methods [5, 29] either neglect the cardinality or regard spatial query area as the cardinality. In fact, we observe from Fig. 1a that a relatively smaller area (left) contains more answer entities than the larger one (right), which is contradicted by the radius of the spatial query area. This leads to sub-optimal query embedding and compromises logical reasoning. Thus, it is necessary to accurately quantify each subset for high-quality query embedding. **Challenge II: How to judiciously leverage the advantage of Guassian distribution?** Multivariate Gaussian distribution has been applied in logical query embedding [9, 29], showing the effectiveness of the expressive parameterization of decision boundaries. Despite some attempts, it is intractable to effectively employ Gaussian distribution for the query embedding with multi-modal distribution due to the following reasons: (1) It typically involves the computation about the covariance matrix (e.g., inverse operation) for the multivariate Gaussian distribution, incurring expensive computational costs. (2) There is no feasible solution to measure the relationships between an entity (Gaussian distribution) and the multiple-answer subsets of a query (Gaussian mixture distribution).

NMP-QEM [29] transfers multivariate Gaussian mixture distribution into a single centroid vector with weighted sum operation and then uses $L_1$ distance to compute the similarity between an entity and a query, which is fundamentally uni-modal distribution modelling for logical reasoning. Hence, it is challenging to effectively employ the Gaussian distribution for modelling multiple disjoint answer subsets.

In this paper, we study the problem of modelling the diversity of answers for logical reasoning over knowledge graphs by addressing the above challenges. We propose Query2GMM, a novel query embedding approach for complex logical query answering. Concretely, we present the GMM embedding to represent each query based on a univariate Gaussian mixture model. Each subset of a query is encoded by its cardinality, semantic center and dispersion degree, elegantly corresponding to the normalized probability, mean value and standard deviation of a univariate Gaussian mixture model. This can offer a more appropriate and precise representation for each subset (**Challenge I**). Additionally, Query2GMM can independently learn these three parameters by using the Cartesian product of a $d$-dimensional univariate Gaussian mixture distribution as the representation of the answer set, which enables Query2GMM fit complex answer sets as well as maintain the linear complexity of the model. Last, we design a new similarity measure (called mixed Wasserstein distance) based on the Wasserstein distance [40], allowing for computing relationships between entities and multiple answer subsets and fitting bidirectional learning for queries and answer entities. The mixed Wasserstein distance forms the foundation of the multi-modal distribution modelling during the reasoning process (**Challenge II**). Extensive experimental results demonstrate that our Query2GMM achieves an average absolute improvement of 6.35% compared to the best baseline.

## 2 PRELIMINARIES

**Knowledge Graphs.** A knowledge graph is denoted as $\mathcal{G} = \{V, R, \Theta\}$, where $V$, $R$ and $\Theta$ refer to the entity set, relation set and fact triplet set, respectively. To record the basic connections of the knowledge graph, a binary relational function: $\mathcal{V} \times \mathcal{V} \rightarrow \{1, 0\}$ is used to memory triplets joined by each $r_j \in R$ with head entity $h_i \in V$ and

tail entity $\theta_k \in V$, where $r_j(h_i, \theta_k) = 1$ if and only if $(h_i, r_j, \theta_k)$ is a factual triple.

**First-Order Logic (FOL).** FOL queries in the literature involve logical operations including existential quantification ($\exists$), conjunction ($\wedge$), disjunction ($\vee$) and negation ($\neg$). We use FOL queries in its Disjunctive Normal Form (DNF) [37], which represents FOL queries as a disjunction of conjunctions for effective computation.

**Computation Graphs.** Following the convention, a logical query is represented by a Directed Acyclic computation Graph (DAG): $Q = \{U, R, L\}$, where $U = \{\tilde{U}, U_?\}$ denotes the node set with $\tilde{U}$ referring to anchor nodes that are source nodes of DAG and $U_?$ denoting variable nodes, $R$ denotes the relation set that is the same as the relation set of $\mathcal{G}$, and $L$ denotes logical operations. Fig. 1b gives an instantiated computation graph of our Query2GMM.

**Problem Statement.** In this paper, we aim to model the ideal distribution of answer entities of logical queries, i.e., multi-modal distribution, along the line of the query embedding approach. Concretely, queries and entities are mapped into the same low-dimensional embedding space, associating each logical operator for entity sets with a transformation in the embedding space. The aim of answering logical queries is to find a set of entities as answers such that these entities' embeddings should be included or close to the embedding of the given query in the embedding space.

## 3 THE PROPOSED QUERY2GMM

### 3.1 GMM embeddings for Entities and Queries

In Query2GMM, we design GMM embedding, i.e., univariate Gaussian mixture model $p(x|\theta) = \sum_{i=1}^{k} \alpha_i \cdot \phi(x|\mu_i, \sigma_i)$ for query embedding, where $\alpha_i$ represents weights of each component satisfying $\sum_{i}^{k} \alpha_i = 1$, $\mu_i$ and $\sigma_i$ stands for mean values of the components respectively, to represent each query. This choice is motivated by the fact that mixed Gaussian distributions can approximate any random distribution arbitrarily well [49]. The cardinality, semantic center, and dispersion degree of each subset of the answer entity set for a query can be intuitively and elegantly represented by the normalized probability, mean value, and standard deviation of a univariate Gaussian mixture distribution, respectively. To better fit complex and diversified answers, we expect that each dimension can be independently learned to explore different information, thus we leverage the Cartesian product for the representation of each sub-distribution [2]. Formally, GMM embedding for a query $q$ is represented as $G_q = [g_1, g_2, \cdots, g_k] \in \mathbb{R}^{k*3d}$, where $k$ is the number of components in the Gaussian mixture distribution. The embedding of each component with $d$-ary Cartesian product of three semantic ingredients can be defined as

$$g_i = g_i^1 \times g_i^2 \times \cdots \times g_i^d = [(\alpha_i^1, \mu_i^1, \sigma_i^1), \cdots, (\alpha_i^d, \mu_i^d, \sigma_i^d)]. \quad (1)$$

Equivalently, Eq. (1) can be formulated as $g_i = [\alpha_i; \mu_i; \sigma_i]$, $i = 1, 2, \cdots, k$. Here, $\alpha_i \in \mathbb{R}^d$ refers to the normalized cardinality proportion of a subset, $\mu_i \in \mathbb{R}^d$ determines the semantic center of a subset and $\sigma_i \in \mathbb{R}^d$ represents the dispersion degree of answers in

a subset. For the complex query answering on large-scale knowledge graphs, there are often one-to-many relationships between entities and queries. Different from existing works that only consider the semantic center for an entity, in this paper, each entity of KGs is represented by a univariate Gaussian distribution embedding with learnable mean value and standard deviation (called Gaussian embedding), i.e., $G_e = [\mu_e; \sigma_e] \in \mathbb{R}^{2d}, \mu_e \in \mathbb{R}^d, \sigma_e \in \mathbb{R}^d$. Here standard deviation serves as the scaling of the distribution, enabling entity embeddings to approach multiple query embeddings with different semantics simultaneously. We also use a univariate Gaussian distribution to represent the relation embedding, i.e., $G_r = [\mu_r; \sigma_r] \in \mathbb{R}^{2d}, \mu_r \in \mathbb{R}^d, \sigma_r \in \mathbb{R}^d$.

Last, for effective computation, we separately learn a trainable transformation matrix for the normalized probability, mean value, and standard deviation. This allows us to formulate the embedding representation of the entity set that corresponds to the provided anchor entity as follows:

$$\begin{aligned} G_q &= [T_\alpha; T_\mu; T_\sigma], \quad T_\alpha = O_\alpha, \\ T_\mu &= \mu_e + O_\mu, \quad T_\sigma = \sigma_e + O_\sigma, \end{aligned} \quad (2)$$

where $O_\alpha \in \mathbb{R}^{k*d}$, $O_\mu \in \mathbb{R}^{k*d}$ and $O_\sigma \in \mathbb{R}^{k*d}$ are the learnable expansion matrices for the normalized probability, mean value, and standard deviation, respectively. Next, we introduce specific neural models of FOL operations, including projection, intersection, negation and union.

### 3.2 Projection

Given a GMM embedding $G_h \in \mathbb{R}^{k*3d}$ of the input head entity set and Gaussian embedding $G_r \in \mathbb{R}^{2d}$ of the input relation, the projection operator aims to obtain the tail entity set connected with any entity in the head entity set with the given relation (see Fig. 2a). Unlike uni-modal distribution learning, it is challenging to learn the overall transformation from the input entity set to the target entity set for relation modelling in the multi-modal distribution context [5, 29]. To address it, we employ a distributed gate [35] to control the different adjustment directions and magnitudes of different subsets of the head entity set based on the input relation, which is formulated as

$$G_{r_{aug}} = [\alpha_{r_{aug}}; G_r], \quad g_t = \sigma(\text{LN}(W_g G_{r_{aug}} + U_g G_h)). \quad (3)$$

$$\begin{aligned} \tilde{G}_h &= \text{ReLU}(\text{LN}(W_h G_{r_{aug}}) + U_h G_h), \\ \tilde{G}_t &= g_t \odot G_h + (1 - g_t) \odot \tilde{G}_h, \end{aligned} \quad (4)$$

where $\alpha_{r_{aug}} \in \mathbb{R}^d$ is the learnable normalized probability vector to formally align the dimension of relation embedding with that of the entity set embedding. $\sigma(\cdot)$ is the sigmoid function, and $g_t$ is the distributed gate to make different adjustments with $k$ subsets with the same relation. $\text{LN}(\cdot)$ is the layer normalization [4], and $W_g, U_g, W_h$ and $U_h$ are parameter matrices. By Eqs. (3) and (4), we get the one-to-one mapping relationship between the tail entity set and head entity set. Then we employ the self-attention network [44] to fine-tune the subsets of the generated final tail entity set based on current relative position, cardinality and dispersion degree:

$$\begin{aligned} G_t &= \text{Attention}(\tilde{G}_t W_Q, \tilde{G}_t W_K, \tilde{G}_t W_V), \\ \text{Attention}(Q, K, V) &= \psi\left(\frac{QK^T}{\sqrt{3d}}\right)V, \end{aligned} \quad (5)$$

---
[2]In our context, we use the terms 'subset' and 'sub-distribution' interchangeably.

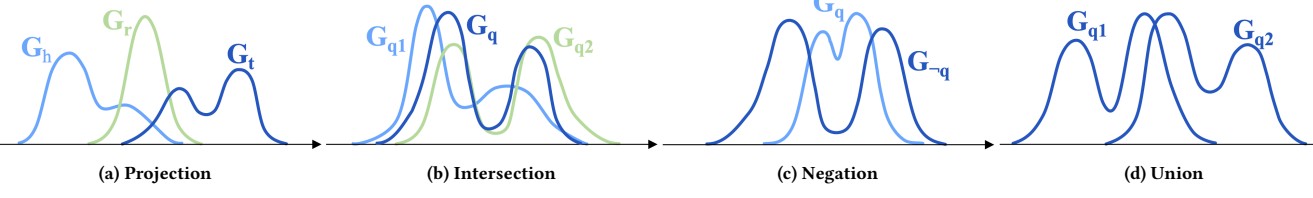

Figure 2: Visualization of logical operator transformation. The input embeddings are represented in light blue and green colors, while the output is shown in deep blue.

where $\psi(\cdot)$ is the Softmax function. This approach allows the $k$ sub-distributions to collaborate, enabling them to cover as many correct answers as possible according to their individual roles.

### 3.3 Intersection

With $m$ GMM embeddings as the input, the intersection operator aims to get the GMM embedding of the target entity set that covers the $m$-intersected answer areas of the input sets (see Fig. 2b). Here "$m$-intersected" indicates that the output of the intersection operator should encompass all $m$ sets of inputs, although they may have varying degrees of overlap (e.g., 2-intersected, 3-intersected). Previous works based on Gaussian distribution [9, 29] handle the "$m-intersection$" problem by sequentially executing a series of pairwise-intersections, which raises several issues below. (1) To obtain the $m-intersected$ answer areas of all $m$ input sets through the pairwise intersection, there are $\frac{n!}{2!}$ possible processing orders and the feasible execution is not unique. Yet, these studies do not provide an optimal execution order or guarantee the permutation invariance of execution orders. (2) The $m-1$ chain executions for the interaction leads to severe cascading errors [13, 16], which negatively impacts the model's performance.

To overcome the above drawbacks, we design a co-attention model to directly derive the "$m-intersected$" areas of $m$ input entity sets. This model can concurrently allocate and aggregate crucial information from both the inter-query level and inter-subset level to fit the candidate entity set.

Concretely, we first compute the overall similarity between $m$ input GMM embeddings and then obtain the inter-query level result using cross attention network:

$$G'_q = \sum_{i=1}^{m} a_i \odot G_{q_i}, \quad a_i = \frac{\exp(MLP(G_{q_i}))}{\sum\limits_{j=1}^{m} \exp(MLP(G_{q_j}))}. \quad (6)$$

By Eq. (6), the output embedding is constrained by existing composition paradigms of the input GMM embeddings. To better encapsulate the interaction among $m*k$ subsets, we employ the self-attention network to capture inter-subset level intersections based on global attention and then utilize the attention pooling [21] to adaptively generate the inter-subset level result. The learning process can be formulated as

$$\tilde{G} = \text{Attention}(GW_Q, GW_K, W_V),$$
$$\text{where } G = [G_{q_1}; G_{q_2}; \cdots; G_{q_m}] \in \mathbb{R}^{(m*k)*3d} \quad (7)$$

$$\text{PoolingAttention}_k(\tilde{G}) = \text{AttentionBlock}(S, rFF(\tilde{G})), \quad (8)$$
$$\text{AttentionBlock}(X, Y) = \text{LN}(H + rFF(H)), \quad (9)$$
$$H = \text{LN}(X + \text{Attention}(XW_Q, YW_K, YW_V)), G''_q = S, \quad (10)$$

where $G \in \mathbb{R}^{(m*k)*3d}$ is the stacked result of the $m$ input embedding and AttentionBlock$(\cdot)$ is a permutation equivariant set attention block network. PoolingAttention$_k(\cdot)$ is a pooling attention network with $k$ seed vectors, which can adaptively aggregate intersection information from the output of self-attention network. $S \in \mathbb{R}^{k*3d}$ is a set of $k$ learnable seed vectors, also, the attention pooling result $G''_q$.

Last, to effectively combine the results of both the inter-query level and the inter-subset level, we employ a gating mechanism to fuse the information of the two parts:

$$g_t = \sigma(W_{g_t}(G'_q \oplus G''_q)), \quad G_q = g_t G'_q + (1 - g_t)G''_q, \quad (11)$$

where $\oplus$ denotes the concatenation of the input matrix. By Eq. (11), we can obtain the final output result by leveraging two-levels interaction learning.

### 3.4 Negation

Given a GMM embedding of an entity set, the aim of the negation operator is to identify the complementary set of the given entity set according to the global cross-correlation among the components of the input entity set (see Fig. 2c). Thus, we utilize a self-attention block layer [44] while excluding the positional encoding, to learn the global correlation among these subsets. We model the negation operator as follows:

$$\tilde{G}_{\neg q} = \text{Attention}(G_q W_Q, G_q W_K, G_q W_V), \quad \tilde{G}_{\neg q} = \text{LN}(G_q + \tilde{G}_{\neg q}), \quad (12)$$

$$G_{\neg q} = \text{LN}(\tilde{G}_{\neg q} + \text{rFF}(\tilde{G}_{\neg q})), \quad (13)$$

where $W_Q \in \mathbb{R}^{3d*3d}, W_K \in \mathbb{R}^{3d*3d}$ and $W_V \in \mathbb{R}^{3d*3d}$ are learnable parameters. rFF$(\cdot)$ refers to a row-wise feedforward layer. By Eq. (12), each subset of the input entity set is aware of the global answer areas. We then encourage the embedding of each sub-distribution to shift towards the areas that are not covered by the given entity set based on Eq. (13).

### 3.5 Union

Given a query $q$, the goal of the union operator is to cover all answer entities from all input entity sets (see Fig. 2d). In Query2GMM, it is natural to model the union of multiple entity sets based on the Gaussian mixture distribution without extra transformation for the

computation graph. Interestingly, based on our initial experiments, we have found that it is more effective to use the DNF technique [37] for union operation. Thus, we transform the union operator to the end of the computation graph and directly use the union of the final multiple solution sets as the output. It is worth noting that the additional cost of query transformation can be alleviated through parallel computation of the conjunctive sub-queries.

## 3.6 Learning GMM Embeddings

**Similarity measurement.** In the reasoning process, we expect that the answer entities should be close to the query. To achieve this, given the Gaussian embedding $G_e = [\boldsymbol{\mu}_e; \boldsymbol{\sigma}_e]$ of an entity and the GMM embedding $G_q = [[\boldsymbol{\alpha}_1; \boldsymbol{\mu}_1; \boldsymbol{\sigma}_1], \cdots, [\boldsymbol{\alpha}_k; \boldsymbol{\mu}_k; \boldsymbol{\sigma}_k]]$ of the target node of the query, we need to measure the similarity between the entity and the query, which serves as the foundation in multi-modal distribution modelling for our Query2GMM.

Fundamentally, we need to measure the similarity between a Gaussian distribution and a Gaussian mixture distribution, but there is no proper similarity metric in the setting of our task. Inspired by [24], we attempt to repeat the Gaussian distribution $k$ times identically to obtain an equally Gaussian mixture distribution, where each dimension in the corresponding GMM embedding for an entity can be represented as $\sum_{i=1}^{k} \frac{1}{k} \cdot \phi(\mu_i, \sigma_i)$. Correspondingly, the original problem can be transformed into the similarity calculation between two Gaussian mixture distributions, and thus existing metrics (e.g., Kullback-Leibler(KL) distance [3, 11] and earth mover's distance(Wasserstein distance) [28, 39, 40]) can be used to measure the similarity between the entity and the query. Here we employ the Wasserstein distance since it satisfies symmetry to measure the similarity between the generated identical Gaussian mixture distribution of the entity and the Gaussian mixture distribution of the query and can fit bidirectional adjustments for queries and answer entities to establish many-to-many mapping relationships.

THEOREM 1. *Given two univariate Gaussian mixture distributions $p_1 = \sum_{i=1}^{m} \alpha_i^1 \phi(\mu_i^1, \sigma_i^1)$ and $p_2 = \sum_{j=1}^{n} \alpha_j^2 \phi(\mu_j^2, \sigma_j^2)$. Let $w_{ij}$ be the Wasserstein distance between Gaussian components $p_{1i}$ and $p_{2j}$. Let $f_{ij}$ be the flow between $p_{1i}$ and $p_{2j}$ that minimizes the overall cost $WORK(p_1, p_2, f) = \sum_{i=1}^{m} \sum_{j=1}^{n} w_{ij} f_{ij}$, which subject to the constraints: (i) $f_{ij} \geq 0$, (ii) $\sum_{i=1}^{m} f_{ij} \leq \alpha_i^1$ for $i \in [1, m]$, (iii) $\sum_{j=1}^{n} f_{ij} \leq \alpha_j^2$ for $j \in [1, n]$, (iv) $\sum_{i=1}^{m} \sum_{j=1}^{n} f_{ij} = \min(\sum_{i=1}^{m} \alpha_i^1, \sum_{j=1}^{n} \alpha_j^2)$. Then, the earth mover's distance(Wasserstein distance) can be defined normalized by the total flow:* $EMD(p_1, p_2) = \frac{\sum_{i=1}^{m} \sum_{j=1}^{n} w_{ij} f_{ij}}{\sum_{i=1}^{m} \sum_{j=1}^{n} f_{ij}}$.

The proof of Theorem 1 is provided in [39]. Next, we elaborate on the definition of the Wasserstein distance (p=2) [53] between a Gaussian distribution $p_e$ and a Gaussian mixture distribution $p_q$ in the setting of Query2GMM. Here, each dimension of both the obtained identical Gaussian mixture embedding of the entity and

the Gaussian mixture embedding of the query satisfies $\sum_{i=1}^{k} \alpha_i = 1$ and then we can obtain $\sum_{i=1}^{k} \sum_{j=1}^{k} f_{ij} = 1$. Meanwhile, each Gaussian component of the corresponding GMM embedding of the entity is identical, we can simplify $EMD(p_1, p_2)$ to $\sum_{j=1}^{k} w_j f_j$. Considering the constraints in Theorem 1, we assume that $f_{ij}$ are identical constants for $\forall i, j$. Below, we define the distance function of the Gaussian distribution $p_e$ and the Gaussian mixture distribution $p_q$ as

$$\mathrm{D}[p_e || p_q] = \sum_{i}^{k} \frac{1}{k} w_i[\phi(\mu_i^e, \sigma_i^e) || \phi(\mu_i^q, \sigma_i^q)], \quad (14)$$

where $w_i[\phi(\mu_i^e, \sigma_i^e) || \phi(\mu_i^q, \sigma_i^q)] = \sqrt{\|\mu_i^e - \mu_i^q\|_2^2 + \|\sigma_i^e - \sigma_i^q\|_2^2}$ is the Wasserstein distance. Nevertheless, Eq. (14) is not directly used for Query2GMM due to the lack of direct measurement and constraints on cardinality. To address it, we attempt to add additional terms on Eq. (14), which is motivated by the KL divergence distance [11] as well as considering the symmetry requirement for the distance function. Hence, we formulate the final distance function as

$$\mathrm{D}[p_e || p_q] = \sum_{i}^{k} \alpha_i^e \log \frac{\alpha_i^e}{\alpha_i^q} + \sum_{i}^{k} \alpha_i^q \log \frac{\alpha_i^q}{\alpha_i^e}$$
$$+ \sum_{i}^{k} \frac{1}{k} w_i[\phi(\mu_i^e, \sigma_i^e) || \phi(\mu_i^q, \sigma_i^q)], \quad (15)$$

where $\alpha_i^e = \frac{1}{k}$. Last, given the Gaussian embedding $G_e$ of an entity and the GMM embedding $G_q$ of a query, the mixed Wasserstein distance between an entity and a query can be defined as the sum of distribution distance along each dimension:

$$\mathrm{D}_{\mathrm{mix}}(G_e; G_q) = \sum_{j=1}^{d} \mathrm{D}[p_{e,j} || p_{q,j}]. \quad (16)$$

Therefore, we can compute the relationships between entities and queries by our proposed mixed Wasserstein distance $\mathrm{D}_{\mathrm{mix}}(\cdot)$ in Eq. (16). This serves as the foundation for multi-modal distribution modelling in logical reasoning.

**Training Objective.** To minimize the distance between the GMM embedding of the query and the Gaussian embeddings of its answer entities, we take the normalized probability of an entity $e$ being the correct answer of the query $q$ by using the Softmax function on all similarity scores. To be specific, we compute the reciprocal of the distance function as the similarity score [53]. Following Query2Particles [5], we also use cross-entropy loss constructed from the given probabilities. The loss function aims to maximize the log probabilities of all correct query-answer pairs:

$$L = -\frac{1}{M} \sum_{t=1}^{M} \log \frac{\exp(1/\mathrm{D}_{\mathrm{mix}}(G_e; G_q))}{\sum_{e' \in V} \exp(1/\mathrm{D}_{\mathrm{mix}}(G_{e'}; G_q))}, \quad (17)$$

where $M$ is the amount of ground-truth answer entities.

**Remarks.** Notably, existing works [9, 29] utilize multivariate Gaussian distribution to learn the query embedding, there are major differences between our Query2GMM and them: (1) Fundamentally, only

**Table 1: MRR results (%) for answering queries on two KGs. $A_p$ and $A_n$ represent the average score of EPFO queries and queries with negation, respectively. The best result is highlighted in bold.**

| KG | Model | Query | | | | | | | | | | | | | | | |
|----|-------|----|----|----|----|----|----|----|----|----|-----|-----|-----|-----|-----|-------|-------|
| | | 1p | 2p | 3p | 2i | 3i | ip | pi | 2u | up | 2in | 3in | inp | pin | pni | $A_p$ | $A_n$ |
| FB237 | BetaE | 38.9 | 10.7 | 9.8 | 29.1 | 42.4 | 12.0 | 22.3 | 11.9 | 9.9 | 4.8 | 7.9 | 7.4 | 3.4 | 3.3 | 20.8 | 5.4 |
| | PERM | 42.4 | 23.9 | 18.6 | 27.5 | 37.6 | 11.8 | 20.3 | 19.4 | 10.7 | - | - | - | - | - | 23.6 | - |
| | NMP-QEM | 44.7 | 17.8 | 13.8 | 32.5 | 43.6 | 11.5 | 22.9 | 17.3 | 12.5 | 5.8 | 9.7 | 7.2 | 4.4 | 3.7 | 24.1 | 6.2 |
| | Q2P | 36.7 | 25.7 | 22.7 | 28.7 | 43.7 | 11.4 | 20.2 | 19.2 | 16.9 | 4.4 | 9.7 | 7.5 | 4.6 | 3.8 | 25.0 | 6.0 |
| | GNN-QE | 42.8 | 14.7 | 11.8 | **38.3** | **54.1** | **31.1** | 18.9 | 16.2 | 13.4 | **10.0** | **16.8** | 9.3 | **7.2** | **7.8** | 26.8 | **10.2** |
| | SConE | 44.2 | 13.0 | 10.7 | 33.8 | 47.0 | 17.0 | **25.1** | 15.5 | 10.7 | 6.9 | 10.6 | 7.9 | 4.0 | 4.3 | 24.1 | 6.7 |
| | WFRE(best) | 44.1 | 13.4 | 11.1 | 35.1 | 50.1 | 27.4 | 17.2 | 13.9 | 10.9 | 6.9 | 11.2 | 8.5 | 5.0 | 4.3 | 24.8 | 7.2 |
| | LMPNN | 45.9 | 13.1 | 10.3 | 34.8 | 48.9 | 17.6 | 22.7 | 13.5 | 10.3 | 8.7 | 12.9 | 7.7 | 4.6 | 5.2 | 24.1 | 7.8 |
| | Ours | **47.3** | **29.3** | **24.2** | 36.8 | 51.5 | 11.1 | 24.3 | **24.8** | **19.2** | 9.6 | 16.3 | **10.3** | 7.1 | 5.3 | **29.9** | 9.7 |
| NELL | BetaE | 53.7 | 13.4 | 11.6 | 37.6 | 48.2 | 15.5 | 24.7 | 12.4 | 9.2 | 5.2 | 7.5 | 10.1 | 3.1 | 3.2 | 25.1 | 5.8 |
| | PERM | 45.2 | 20.4 | 17.7 | 29.6 | 43.8 | 12.2 | 17.8 | 30.2 | 18.0 | - | - | - | - | - | 26.1 | - |
| | NMP-QEM | 52.2 | 23.7 | 18.9 | 33.4 | 44.1 | 15.8 | 20.9 | 22.8 | 13.7 | 4.8 | 8.2 | 9.3 | 3.8 | 3.5 | 27.3 | 5.9 |
| | Q2P | 54.3 | 28.8 | 29.3 | 29.4 | 45.4 | 10.9 | 18.1 | 36.8 | 19.5 | 5.1 | 7.4 | 10.2 | 3.3 | 3.4 | 30.3 | 6.0 |
| | GNN-QE | 53.3 | 18.9 | 14.9 | **42.4** | 52.5 | **30.8** | 18.9 | 15.9 | 12.6 | 9.9 | **14.6** | 11.4 | **6.3** | 6.3 | 28.9 | 9.7 |
| | SConE | 58.2 | 20.5 | 17.0 | 41.8 | 50.7 | 22.9 | **28.6** | 18.8 | 15.5 | 6.2 | 8.0 | 11.8 | 3.5 | 4.2 | 30.4 | 6.7 |
| | WFRE(best) | 58.6 | 18.6 | 16.0 | 41.2 | 52.7 | 28.4 | 20.7 | 16.1 | 13.2 | 6.9 | 8.8 | 12.5 | 4.1 | 4.4 | 29.5 | 7.3 |
| | LMPNN | 60.6 | 22.1 | 17.5 | 40.1 | 50.3 | 24.9 | 28.4 | 17.2 | 15.7 | 8.5 | 10.8 | 12.2 | 3.9 | 4.8 | 30.7 | 8.0 |
| | Ours | **64.1** | **36.6** | **34.3** | 40.8 | **55.7** | 17.1 | 26.7 | **46.7** | **22.4** | **10.0** | 13.9 | **15.3** | 6.3 | **6.7** | **38.3** | **10.4** |

Query2GMM is capable of learning diversified answers with multi-modal distribution in the reasoning process. This is done through our design and the proposed mixed Wasserstein distance. However, existing works do not. (2) Different from them, Query2GMM takes the Cartesian product for the embedding representation based on univariate Gaussian (mixture) distribution. This maintains the linear complexity of the model and avoids complicated computation, e.g., the inverse of the covariance matrix. (3) We design more effective neural models for logical operators considering their own properties in the multi-modal distribution context. This helps handle complex mapping relationships and alleviates cascading errors.

## 4 EXPERIMENTS

### 4.1 Experimental Settings

**Datasets.** We evaluate the proposed Query2GMM on two KG benchmark datasets, i.e., FB15k-237(FB237) [43] and NELL995(NELL) [50]. FB15k-237 is a subset of FB15k [7] which removes the inverse relation and avoids the issue of data leakage [12] criticized by FB15k. We split the edges of each original graph into training edges, validation edges, and test edges. And the corresponding training graph $\mathcal{G}_{training}$, validation graph $\mathcal{G}_{validation}$ and test graph $\mathcal{G}_{test}$ satisfies $\mathcal{G}_{training} \subseteq \mathcal{G}_{validation} \subseteq \mathcal{G}_{test}$.

**Baselines.** We compare our proposed Query2GMM with eight existing approaches, including (i) Multi-modal based solutions: NMP-QEM [29] and Query2Particles(Q2P) [5] follow the multi-modal preset, in which NMP-QEM is based on distribution embedding; (ii) Uni-modal based solutions: BetaE [38], PERM [9] and SConE [33], where the first two methods leverage probabilistic distribution to generate query embedding and the last one designs geometric embedding to represent complex logical queries. (iii) Other embedding

based solutions: GNN-QE [55], WFRE [47] and LMPNN [48] attempt to employ different kinds of learning paradigm for logical query answering, such as pre-training techniques [48] and classical link prediction models [55].

**Training Protocol.** We implement Query2GMM in Pytorch on Nvidia RTX 3090 GPU. We set embedding dimension $d$ as 400 and use the uniform distribution for the parameter initialization. Additionally, we utilize grid search for hyperparameter tuning, where the learning rate varies from 0.0001 to 0.001. For the batch size, we set 2048 and 1024 for the FB237 dataset and NELL dataset, respectively. We optimize the loss function in Eq. (17) using AdamW optimizer [30].

### 4.2 Reasoning over Knowledge Graphs

To evaluate the quality of query embedding, we compare our Query2GMM with eight baselines. Following the setting of baselines [5, 38], we use fourteen types of queries based on the first-order logic queries and report Mean Reciprocal Rank (MRR) results on FB237 and NELL. Note that we obtain results of GNN-QE and WFRE directly from their original papers due to the limitations of GPU resources and report results of other baselines by re-running their models based on the configuration explained in their papers.

We can observe from Table 1 that (1) Our Query2GMM significantly outperforms all baselines in most cases. This underlines the effectiveness of our proposed embedding backbone (i.e., GMM embedding), coupled with our neural operators, in accurately representing complex logical queries. (2) Multi-modal based models (NMP-QEM, Q2P, and Query2GMM) demonstrate better empirical performance compared to uni-modal based models (BetaE, PERM,

**Table 2: Ablation study on NELL regarding MRR score. The best result is highlighted in bold.**

| Method | Query | | | | | | | | | |
|---|---|---|---|---|---|---|---|---|---|---|
| | 1p | 2p | 3p | 2i | 3i | ip | pi | 2u | up | Average |
| Query2GMM | **64.1** | **36.6** | **34.3** | **40.8** | **55.7** | **17.1** | **26.7** | **46.7** | **23.4** | **38.3** |
| w/o Cardinality | 60.2 | 33.8 | 32.2 | 35.2 | 51.1 | 14.2 | 22.1 | 41.6 | 19.2 | 34.4 |
| w/o Dispersion | 60.7 | 35.0 | 31.7 | 36.2 | 52.3 | 14.8 | 22.7 | 41.2 | 19.5 | 34.9 |
| Query2GMM$_{pro}$ | 58.7 | 32.7 | 31.4 | - | - | 7.4 | 18.1 | - | 19.2 | - |
| Query2GMM$_{inter}$ | - | - | - | 32.6 | 42.5 | 14.2 | 19.4 | - | - | - |
| Query2GMM$_{mWD}$ | 63.2 | 35.6 | 33.8 | 36.2 | 52.5 | 15.9 | 23.5 | 43.4 | 21.3 | 36.2 |

and SConE), implying that they are more adept at fitting diverse answers. (3) Compared to NMP-QEM, which uses a multivariate Gaussian mixture distribution for the query embedding, Query2GMM shows an overall average improvement of 12.4%. This is mainly because the proposed mixed distribution of NMP-QEM degenerates to a single distribution with the distance evaluation method during the model reasoning process, despite its initial structure following a multi-modal distribution. This emphasizes the crucial role of reasoning within the multi-modal distribution learning framework. Unlike the inappropriate distance function in NMP-QEM that neglects the characteristics of multi-modal distribution, our proposed mixed Wasserstein distance guarantees effective implementation of intricate logical reasoning based on multi-modal distribution preset. (4) The performance of complex logical queries can be enhanced by accurately representing multiple subsets, as shown in the comparison between Query2GMM and Q2P. We will provide more detailed benefits in the following subsection. (5) Although LMPNN and GNN-QE leverage advanced techniques (i.e., pre-training and link prediction), our Query2GMM yields superior performance in most cases compared to these two baselines, demonstrating the effectiveness of multi-subset modeling in answering complex logical queries. Additionally, we observe that Query2GMM underperforms on certain queries involving the intersection operator. This is primarily due to the increased complexity associated with intersection relationships in the context of multi-modal distributions, which scale by a factor of $k$. However, by considering two-level intersections, we manage to alleviate this issue. As a result, Query2GMM outperforms multi-modal based baselines on logical queries with the intersection operator.

## 4.3 Ablation Study

In the ablation study, we conduct extensive experiments to justify the effects of five key components of Query2GMM, including (1) the cardinality of GMM embedding, (2) the dispersion degree of GMM embedding, (3) the proposed neural projection operator, (4) the proposed neural intersection operator, and (5) the proposed mixed Wasserstein distance $D_{mix}(\cdot)$. Here, all experiments are conducted on queries on the NELL dataset in terms of MRR.

**GMM embedding.** To justify the effect of our GMM embedding, we conduct two variants "w/o Cardinality" and "w/o Dispersion" by removing the cardinality parameter and dispersion degree parameter, respectively. As shown in Table 2, cardinality and dispersion degree play individual and important roles in accurate representation for

query embedding in the multi-modal distribution context, achieving an average improvement of 3.9% and 3.4%, respectively. This is because w/o Cardinality fails to provide prompts for the number of answer entities during the reasoning process and Query2GMM without Dispersion compromises the ability of joint learning for multiple answering subsets of complex logical queries. In summary, this suggests that the effectiveness of multi-modal distribution modeling for logical queries relies on three elements: cardinality, semantic center, and degree of dispersion.

**Projection operation.** To evaluate the effectiveness of our proposed projection operator, we replace our gated attention network with 3-layer MLPs, called Query2GMM$_{pro}$. As we can see from Table 2, the performance of Query2GMM$_{pro}$ drops up to 9.7%, affirming the effectiveness of our proposed model: independent component learning based on gate mechanism, in conjunction with joint learning using the self-attention mechanism.

**Intersection operation.** To show the effectiveness of the proposed co-attention network for the intersection operator, we create a variant: Query2GMM$_{inter}$, by replacing our neural model with self-attention and random sampling used in Q2P [5] for the intersection operator. We can observe from Table 2 that the performance of the variant Query2GMM$_{inter}$ falls by up to 13.2%, showing that effectively modeling at the inter-query and inter-subset levels can more accurately adapt to the complex mapping relationships inherent in intersection operations. Meanwhile, our proposed co-attention model with the two-level relationship learning can adaptively generate results, thereby facilitating the coverage of $m$-intersected answer areas in the multi-modal distribution context.

**Mixed Wasserstein distance.** To illustrate the effectiveness of our designed mixed Wasserstein distance, we introduce a variant Query2GMM$_{mWD}$, i.e., using the Eq.(14) for distance computation, which neglects the cardinality part of GMM embeddings. Observe from Table 2 that the performance of Query2GMM$_{mWD}$ reduces by up to 4.6%, with an average decline of 2.1% compared to Query2GMM. This highlights that direct constraints on the cardinality, semantic center, and dispersion degree of GMM embeddings within the loss function aid in superior model learning for logical reasoning. Additionally, Query2GMM$_{mWD}$ consistently outperforms "w/o Cardinality", suggesting that the cardinality part of our GMM embedding plays a critical role in the model's reasoning in the multi-modal setting, although Query2GMM$_{mWD}$ only implicitly learns the cardinality of GMM embedding. In summary, this underpins the effectiveness of our GMM embedding design.

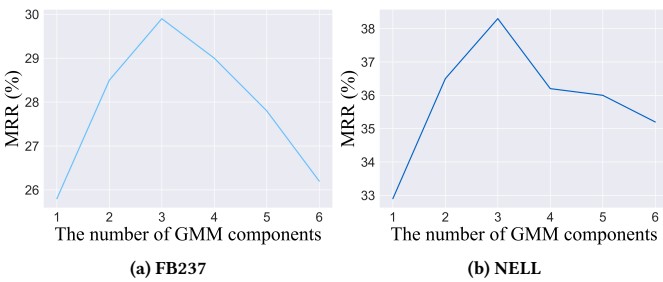

**Figure 3: Comparison of average MRR scores on EPFO queries across two datasets using different GMM components.**

## 4.4 Parameter Analysis

In this section, we further study the effect of an important parameter in Query2GMM, that is, the number of Gaussian components $k$ in GMM embedding. We construct experiments by varying $k$ from 1 to 6 on both FB237 and NELL datasets and report average MRR scores on EPFO [37] queries. Observe from Fig. 3 that (i) Multimodal distribution modeling ($k > 1$) is better than uni-modal based representation ($k = 1$), which supports the fundamental premise of our model. Moreover, multiple components for each query can help effectively reduce false positives, resulting in a more precise representation of the answer areas for each query. (ii) Query2GMM achieves the best performance when using a smaller number of Gaussian distribution components, specifically for $k = 3$. However, as $k$ increases from 3 to 6, Query2GMM shows a performance decrease since this makes the joint learning of these components more challenging as well as increases the risk of overfitting. Additionally, (iii) on the NELL dataset, Query2GMM with $k \in [1, 6]$ consistently outperforms the best competitor among eight baselines, demonstrating the overall effectiveness and robustness of our model design, including the design of GMM embedding and corresponding operator networks.

## 5 RELATED WORKS

Answering complex logical queries is one of the long-standing topics in reasoning on knowledge graphs. The existing works can be broadly grouped into subgraph-based and embedding-based methods. Specifically, the former [15, 26, 31, 42] employs classical subgraph matching algorithms to search top-$K$ similar subgraphs in the knowledge graph as the answers, given a directed acyclic query graph corresponding to a query. Unfortunately, these solutions cannot find correct answers from incomplete/noisy knowledge graphs [14, 34], which are common in practice. Moreover, their response time often fails to meet real-time requirements, leading to the prolonged delay [25].

To address the challenges for real-life KG reasoning, much more attention is put on embedding-based methods, in which various advanced deep learning techniques are applied to achieve promising performance in answering logical queries over incomplete KGs. Along this line, most existing methods can be classified into two groups: uni-modal based models and multi-modal based models. In the former group, several geometric embeddings [10, 17, 25, 32,

37, 54] were proposed to represent logical queries and entities, and then answering entities corresponding to each query can be simply obtained in the low-dimensional space, after which symbolic reasoning is integrated into geometry learning to improve the interpretability of models [33]. To model a complete set of first-order logical operations, BetaE [38] proposed to leverage probability distribution to fit logical queries, where each embedding consists of multiple independent Beta distributions to capture different aspects of a given entity or a query. Based on this idea, LinE [19] extended probabilistic embedding to the Line embedding space to improve the model expressivity. However, the above methods based on a strong assumption inevitably introduces more false positive answers, violating the semantics of query embeddings and then degrading the performance, as it is validated in the literature [5, 29] and our experiments.

To accurately represent the ideal distribution of answer entities, multi-modal based methods [5, 29] were presented to model the multiple subsets of each query in the embedding space. Concretely, Query2particles [5] putted forward to use multiple vectors to characterize the diverse candidate answers of a complex KG query based on the assumption of multi-modal distribution. NMP-QEM [29] proposed the idea of encoding the answer set of each mini-query of a complex logical query using mixed multivariate Gaussian distribution. However, it evaluates the relationships between an entity and multiple subsets of a query inappropriately, leading to multi-modal distribution modeling degenerating into uni-modal distribution modeling during the reasoning process. Therefore, our research endeavors to explore an accurate subset representation of logical queries, which still remains an open problem to be resolved.

There are other explorations in logical reasoning. Researchers [6, 8, 41, 46, 48, 55] shifted their attention to the query graph, where complicated queries were first decomposed into different patterns based on different techniques including tree decomposition and fuzzy logic. Besides, pre-training techniques were extensively applied to complete logical queries on knowledge graphs [2, 18, 27], which can help improve transferability and generability. However, our experiments also validate the effectiveness of Query2GMM compared to these methods.

## 6 CONCLUSION

In this paper, we investigate the multi-modal distribution of answers for each query in logical reasoning over knowledge graphs. To achieve this, we propose Query2GMM, a query embedding method for complex logical query answering. Unlike existing works, Query2GMM is able to elegantly and accurately represent each subset using cardinality, semantic center and dispersion degree with the proposed GMM embedding. Furthermore, we design a new mixed Wasserstein distance to measure the relationships between an entity and multiple answer subsets of a query. This provides a solid foundation for reasoning within multi-modal distribution learning. Extensive experiments demonstrate the effectiveness of our proposed Query2GMM and each of its key components. Crucially, we empirically observe that modelling the multi-modal distribution during the reasoning process is of greater importance than the initial representation for complex logical query answering.

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
