# OpenReview forum: "Query2GMM: Learning Representation with Gaussian Mixture Model for Reasoning over Knowledge Graphs"
_ACM.org/TheWebConf/2024/Conference — TheWebConf24 Oral_

### Official Review · Reviewer_oq1Q · 2023-11-19

**Novelty:** 4
**Technical Quality:** 5

**Review:**

This paper improved the multi-modal reasoning process for the logical query answering problem over knowledge graphs, by modeling the multi-modal reasoning process via more accurate Gaussian Mixture model as well as designing new multi-modal distance representations. At the same time, a more effective neural model for logical operators is proposed, taking a simple improvement on the previous work, and proving the effectiveness of its components through extensive experiments. The overall writing is clear and logical, the experimental results obtained are optimal in most cases, and the quality of the work is good.

However, there are some problems which should be addressed before it is considered for acceptance. If the following problems are well-addressed, I believe that the essential contributions of this paper are important for logical query answering over knowledge graphs.

The role of cardinality is not adequately demonstrated in the paper, and even though the experimental part of the ablation experiment demonstrates its effectiveness, it does not explain the principle of its action well enough, and there should be more experiments beyond the modeling effect to justify this "elegant" Gaussian Mixture representation.

Some of the sentences discussing the role and problems of components are not concise enough and the content is repetitive and redundant.

The work improves on the previous work on multi-modal Gaussian Mixture query representation, which is not as innovative as it could be.

**Questions:**

Please explain the mechanism of cardinality's role in elegantly modeling multi-modal Gaussian mixture distribution learning for reasoning, and the role of the new mixed Wasserstein distance designed to accurately model the reasoning process, thus demonstrating that the article's improvements to the previous work on multivariate Gaussian distribution are important and essential.

Please explain exactly how the baseline for the ablation experiments on neural models for logical operators was designed. More importantly, why not use a similar connection to the normal residual network as a comparison baseline for ablation experiments?

**Reviewer Confidence:**

2: The reviewer is willing to defend the evaluation, but it is likely that the reviewer did not understand parts of the paper

**Scope:**

3: The work is somewhat relevant to the Web and to the track, and is of narrow interest to a sub-community

---

### Official Review · Reviewer_MSrd · 2023-11-23

**Novelty:** 5
**Technical Quality:** 4

**Review:**

*By mistake, I submitted a wrong review to this paper, I now replaced it with the correct one. Apologies for the confusion*

This paper presents a method for query answering over knowledge graph by embedding both queries and KG entities in the same embedding space. The main innovation of the method presented in the paper is the use of multi-modal distributions in the form of a Gaussian Mixture model. The method takes as input a KG and First-Order Logical formulae that express queries. For each of the logical operations (negation, conjunction etc), an embedding submethod is defined. The combination of said methods lead to a GMM distribution in the embedding space. A Wasserstein Distance metric is then used in the learning procedure.

The paper presents the method and evaluation on two datasets, comparing it to several baseline methods. The method mostly outperforms the state of the art. Furthermore, ablation studies are presented that show the influence of the different elements of the method.

The paper the method in a mostly mathematical manner but also is able to convey the main intuitions well. I have a few remaining questions and remarks regarding the paper.

- The related work section is now presented at the end. I would suggest to move it to the front of the paper. several questions I had while reading the paper are addressed only in the related work section.

- One of these questions concerns by to do this using statistical methods at all, why not do (logical) pattern matching. The related work section states that this is not adequate for incomplete knowledge, but this claim is not empirically backed up. How do said methods perform on the queries in the experimental setup? Also, such methods are claimed to not provide answers in real-time, but again, that claim is not backed up by empirical results in the paper. It would be good to compare the learning/query response time of the Query2GMM with such logical pattern matching methods.

- Regarding empirical evaluation, shown in Table 1. Where do these queries come from (1p, 2p etc)? What characteristics do they have? To what extent do they use unknown information? This is not made clear in the paper and makes it more difficult to assess the usefulness of the solution.

**Questions:**

- The related work section is now presented at the end. I would suggest to move it to the front of the paper. several questions I had while reading the paper are addressed only in the related work section.

- One of these questions concerns by to do this using statistical methods at all, why not do (logical) pattern matching. The related work section states that this is not adequate for incomplete knowledge, but this claim is not empirically backed up. How do said methods perform on the queries in the experimental setup? Also, such methods are claimed to not provide answers in real-time, but again, that claim is not backed up by empirical results in the paper. It would be good to compare the learning/query response time of the Query2GMM with such logical pattern matching methods.

- Regarding empirical evaluation, shown in Table 1. Where do these queries come from (1p, 2p etc)? What characteristics do they have? To what extent do they use unknown information? This is not made clear in the paper and makes it more difficult to assess the usefulness of the solution.

**Ethics Review Description:**

-

**Reviewer Confidence:**

3: The reviewer is confident but not certain that the evaluation is correct

**Scope:**

3: The work is somewhat relevant to the Web and to the track, and is of narrow interest to a sub-community

---

### Official Review · Reviewer_6CnQ · 2023-11-24

**Novelty:** 6
**Technical Quality:** 5

**Review:**

The paper addresses the general idea of using machine learning to provide approximate answers to graph queries.  The general idea is to learn vector space embeddings of both the entities in the knowledge graph and the queries, and then to which entities are correct answers using an operation on the two embeddings.

One basic idea is to use embeddings that represent geometric regions in the embedding space, where the answers are the entities whose embeddings fall into that region.  This does not work very well when the shapes of the regions are too simple.

This paper suggests embedding queries into a gaussian mixture distribution, i.e. a vector containing the parameters of such a distribution.  The multi-modal nature of this embedding makes it better suited for representing the answer sets of complex queries, e.g. containing unions.

The work seems sound to me, but I am not an expert.  I have some questions.

**Questions:**

1. The paper mentions "reasoning" a lot, but as far as I can see, the approach addresses first-order queries, i.e. queries that amount to model checking of first order formulae.  This is the same as SQL queries over relational data for instance.  This is not usually referred to as reasoning in this community, it is simply query evaluation.  We talk about reasoning when there is additionally a TBox (e.g. OWL ontology) and the answers to a query under the combination of the the instance data in the graph and the axioms in the TBox are required.  I think this is not happening in this work?

1. The most common approach to answering queries is usually fast, gives 100% exact answers and does not involve machine learning at all.  I’m missing that baseline… the approach certainly does not outperform usual query processing in terms of accuracy. It it faster?  Does it use less of another resource?  When taking into account the learning phase?

1. The usual approach to query processing, after deciding a query plan, is to evaluate basic graph patterns (joins of triples), then combining them using union, projection, optional, etc.  How does your approach compare to using ML techniques for the basic graph patterns, and combining the results using conventional methods?

**Reviewer Confidence:**

2: The reviewer is willing to defend the evaluation, but it is likely that the reviewer did not understand parts of the paper

**Scope:**

4: The work is relevant to the Web and to the track, and is of broad interest to the community

---

### Official Review · Reviewer_9nC5 · 2023-11-25

**Novelty:** 6
**Technical Quality:** 4

**Review:**

This work introduces Query2GMM, a Gaussian Mixture Model-based approach for answering logical queries over knowledge graphs. Distinct from existing multi-modal works, Query2GMM is capable of quantifying different answer subsets for each query, as it encodes the cardinality in the representation of a set of entities.

Additionally, Query2GMM incorporates a novel distribution similarity measure. Both an experiment and an ablation study confirm its effectiveness.

Pros:
1. Great presentation, novelty
2. Thorough experimentation

Cons:
1. The problem statement lacks clarity, particularly in the definition of training and test data.
2. There are errors in representation:
2.1 The mention of $rFF(\cdot)$ on page 4, right column, line 453, section 3.4, should be introduced earlier in section 3.3. Furthermore, the representation of $rFF(\cdot)$ in Formulas 8-9 differs from that in Formula 13. The same inconsistency is observed with $MLP(\cdot)$ in Formula 6.
2.2 In Formula 17, the variable $t$ is not defined. Although it is ostensibly the index of the sample, the definition of the sample itself is missing.

**Questions:**

1. How to obtain the union of two sets of entities in Query2GMM?
2. A limitation of the neural models of First-Order Logic (FOL) operations in this paper, is their lack of interpretability and uncertainty in preserving the inherent properties of these FOL operations. Specifically, it remains unclear whether, as the loss function nears zero, the trained parameters can maintain key properties of FOL operators, such as De Morgan's Laws, Distributive Laws, Associative Laws, etc. For two GMM embeddings $G_{q_1}$,$G_{q_2}$ trained on FB237/NELL, what is the mathematic expectation of distribution distance between $Negation(Intersection(G_{q_1},G_{q_2}))$ and $Union(Negation(G_{q_1}), Negation(G_{q_2}))$?
3. Why were experiments not conducted on the WN18RR dataset?

**Reviewer Confidence:**

3: The reviewer is confident but not certain that the evaluation is correct

**Scope:**

3: The work is somewhat relevant to the Web and to the track, and is of narrow interest to a sub-community

---

### Official Review · Reviewer_k9Sh · 2023-11-28

**Novelty:** 6
**Technical Quality:** 6

**Review:**

**Summary:**

The present paper presents Query2GMM, a method using Gaussian mixture models for logical query answering over knowledge graphs.
This paper is inspired by the recent finding that ideal query embeddings might follow a multi-modal distribution. The paper addresses this finding by presenting a method based on GMMs. The method is evaluated on two standard benchmark datasets for the task against a variety of state-of-the-art baselines. On NELL all baselines are clearly outperformed, on FB15K-237, the improvements are smaller, particularly on queries with negation.

**Review:**

The paper presents a novel and technically valuable new method for answering FOL queries on knowledge graphs. The method is well-motivated and well-grounded in recent literature in the field. The presentation of the work is good. The final experimental results are informative, especially because of the extensive ablation studies demonstrating the relevance of the different operators.
Overall, this is a very good paper.


**Strengths:**

-	Novel and very interesting new method for logical query answering
-	Very good and recent related work
-	Strong experimental results
-	Well-written paper

**Questions:**

-	Since the paper does not contain a paragraph on future work, I would be interested in the next steps and shortcomings of the current method.

**Reviewer Confidence:**

2: The reviewer is willing to defend the evaluation, but it is likely that the reviewer did not understand parts of the paper

**Scope:**

4: The work is relevant to the Web and to the track, and is of broad interest to the community

---

### Decision · Program_Chairs · 2024-01-22

**Decision:**

Accept (Oral)

**Comment:**

This article introduces an approach to enable query answering over knowledge graphs through embedding techniques.
 Results show that this approach mostly outperforms exsting approaches.

 All reviewers agree that this work is novel and produces valuable insights and research contributions, and deserves to be accepted.
 We do recommend the reviewers to include the suggested changes from the reviewers, such as ideas on future work and fixes to various minor errors.